# Loss of Cell-Cell Contact Inhibits Cellular Differentiation of α-Catenin Knock Out P19 Embryonal Carcinoma Cells and Their Colonization into the Developing Mouse Embryos

**DOI:** 10.3390/biotech13040041

**Published:** 2024-10-03

**Authors:** Masahiro Sato, Emi Inada, Naoko Kubota, Masayuki Ozawa

**Affiliations:** 1Section of Gene Expression Regulation, Frontier Science Research Center, Kagoshima University, Kagoshima 890-8544, Japan; 2Department of Pediatric Dentistry, Graduate School of Medical and Dental Sciences, Kagoshima University, Kagoshima 890-8544, Japan; inada@dent.kagoshima-u.ac.jp (E.I.); kubotarecta@gmail.com (N.K.); 3Department of Biochemistry and Molecular Biology, Graduate School of Medical and Dental Sciences, Kagoshima University, Kagoshima 890-8544, Japan; masayukiozawa0714@gmail.com

**Keywords:** cell–cell contact, cadherin−catenin complex, embryoid body, cellular differentiation, P19 cells, embryonal carcinoma cell, CRISPR/Cas9, aggregation chimera

## Abstract

Cadherin−catenin cell−cell adhesion complexes, composed of cadherin, β-catenin or plakoglobin, and α-catenin (α-cat) molecules, are crucial for maintaining cell−cell contact and are commonly referred to as “adherens junctions (AJs).” Inactivating this system leads to loss of cell−cell contact and developmental arrest in early embryos. However, it remains unclear whether the loss of cell−cell contact affects the differentiation of embryonic cells. In this study, we explored the use of a murine embryonal carcinoma cell line, P19, as an in vitro model for early embryogenesis. P19 cells easily form embryoid bodies (EBs) and are susceptible to cellular differentiation in response to retinoic acid (RA) and teratoma formation. Using CRISPR/Cas9 technology to disrupt the endogenous *α-cat* gene in P19 cells, we generated *α-cat* knockout (KO) cells that exhibited a loss of cell−cell contact. When cultivated on non-coated dishes, these *α-cat* KO cells formed EBs, but their structures were labile. In the RA-containing medium, the *α-cat* KO EBs failed to produce differentiated cells on their outer layer and continued to express SSEA-1, an antigen specific to pluripotent cells. Teratoma formation assays revealed an absence of overt differentiated cells in tumors derived from *α-cat* KO P19 cells. Aggregation assays revealed the inability of the KO cells to colonize into the zona pellucida-denuded 8-cell embryos. These findings suggest that the AJs are essential for promoting the early stages of cellular differentiation and for the colonization of early-developing embryos.

## 1. Introduction

Cells of “simple epithelia” are connected to each other via a junctional complex, which is comprised by a tight junction, known as an adherens junction (AJ; also called zonula adherens), and a desmosome at the apical-most end of cell–cell contacts [1]. E-cadherin is a main adhesion receptor at the AJ of epithelial cells, which also functions at cell–cell contacts. It binds β-catenin or plakoglobin and in turn α-catenin (α-cat; also called αE-catenin), forming the cadherin−catenin cell−cell adhesion complex [1,2]. α-cat interacts with the actin filament directly or indirectly via binding to vinculin and acts as a mechanosensor [3,4,5]. Since ablation of α-cat resulted in disruption of the cadherin-based AJs, close interaction between the cadherin–catenin cell−cell adhesion complex and actin filament is crucial for establishing cell–cell contacts (Figure 1A; [6]).

The cadherin–catenin cell–cell adhesion complex plays an essential role in early embryogenesis in mammals. For instance, *E-cadherin* knockout (KO) embryos created through gene targeting exhibit developmental arrest at the blastocyst stage, where formation of the trophectoderm, the first epithelial tissue to differentiate in the mouse embryo, is blocked [7,8]. A lack of E-cadherin also causes cellular dysfunction. For instance, inhibition of E-cadherin expression in mouse embryonic stem (ES) cells prevents normal embryoid body (EB) formation and subsequent germ layer differentiation [9]. A human colon cancer cell DLD-1 lacking E-cadherin expression showed a round fibroblast-like morphology. However, the introduction of *E-cadherin* gene (also called *Cdh1*) into these cells results in epithelial cell morphology [10].

Mouse embryos lacking *α-cat* (also called *Ctnna1*) expression also exhibited developmental arrest at blastocysts, like *E-cadherin* KO embryos. For example, Torres et al. [11] constructed trap ES clones, in which a trap vector had been inserted into the exon of endogenous *α-cat* gene, and then produced chimeric mice using aggregation approach between the trap ES cells and morulae. Mating of these offspring revealed that the development of homozygous embryos was blocked at the blastocyst stage. In abnormal embryos, a mass of rounded cells remains inside the zona pellucida (ZP). Interestingly, Torres et al. [11] attempted to cultivate abnormal embryos and found that they divided normally but remained round and did not adhere to one another. In contrast, cultivation of normal embryos resulted in the generation of ES cells with tight cell–cell contact. The property of reduced cell–cell contact and morphological alteration from epitherial to round shape in the *α-cat* KO cells is also seen in other types of differentiated cells such as Type II Madin-Darby canine kidney (MDCK) cells [12] and DLD-1 [13].

Our main concern was how ES cells lacking *α-cat* expression could differentiate when placed under appropriate conditions, allowing cellular differentiation (such as EB formation after cultivation of cells in non-coated dishes or forced induction of cellular differentiation in response to retinoic acid (RA), a potent inducer of cellular differentiation). In other words, pluripotent cells can differentiate in the absence of cell–cell contact.

In this study, we used a mouse P19 embryonal carcinoma (EC) cell line derived from teratocarcinoma in C3H/He mice, produced by grafting an embryo at 7 days of gestation to the testes of an adult male mouse [14,15]. This line has the potentiality to differentiate into several types of differentiated cells (whose property is known as ”multi-differential potentiality”) when they are cultivated under the conditions allowing cellular differentiation (i.e., cultivation in medium containing RA) [16,17,18,19]. Moreover, P19 cells participate in embryogenesis by forming chimeric mice when injected into the blastocoel of a mouse blastocyst. The resulting chimeras had a broad range of cell types derived from the introduced P19 cells [20]. We employed the CRISPR/Cas9 method to disrupt the *α-cat* gene in P19 cells. Using the *α-cat* KO cells, we examined several properties, such as cell morphology, expression of cell-surface marker antigen, susceptibility to cellular differentiation in response to RA, in vivo differentiation ability through an in vivo teratoma formation assay, and the potential to colonize into early embryos, through EC cells/8-cell embryo aggregation assay.

## 2. Materials and Methods

### 2.1. Animals

B6C3F1 female mice (a hybrid of C57BL/6N and C3H/HeN), 7 weeks old, were purchased from Japan SLC, Inc. (Hamamatsu, Shizuoka, Japan) and served as donors for the production of aggregation chimeras between EC and 8-cell embryos. To obtain pregnant females, these B6C3F1 mice were naturally mated with C57BL/6 male mice (8–15 weeks old; Japan SLC, Inc., Shizuoka, Japan). For all matings, any vaginal plugs observed was designated as E0.5. For tumor cell inoculation into the pancreas, 8–20-week-old immunodeficient female mice (BALB/c-nu/nu; CLEA Japan Ltd., Tokyo, Japan) were used. All mice were maintained on a 12-h light/dark schedule (lights on from 07:00 to 19:00) and were allowed food and water ad libitum.

Surgeries were performed under anesthesia using a combination of medetomidine, midazolam, and butorphanol. All efforts were made to reduce the number of animals used and minimize suffering.

### 2.2. Isolation of Cells Lines Using CRISPR/Cas9 Plasmid-Mediated Gene KO

P19 cells were obtained from Prof. Hiroshi Hamada (Osaka University). P19 cells and their derivatives were cultured in DMEM-high glucose (#11995; Invitrogen Co., Carlsbad, CA, USA) containing 10% fetal bovine serum (FBS), referred to as DMEM-high glucose/10% FBS, and 1% antibiotic-antimycotic solution (#A5955; Sigma-Aldrich Co., Ltd., St. Louis, MO, USA).

For the CRISPR/Cas9-mediated KO of genes, we used the pCGSapI vector developed by Takayuki Sakurai (Shinshu University, Nagano, Japan; [21]). This vector contains the human *Cas9* gene under the control of the chicken β-actin promoter-based *CAG* promoter and a unique cloning site, *Sap* I, for inserting the guide RNA (gRNA) under the control of the *U6* promoter. All synthetic oligonucleotides corresponding to the gRNA and its complementary strand contained the adaptor sequence of *Sap* I. The following oligonucleotides were used to construct gRNAs (lowercase letters represent adaptor sequences): 5′-accgTCTGGCTGTTGAAAGACTGTg-3′ and 5′-aaacACAGTCTTTCAACAGCCAGACc-3′. The sequence, recognized by the gRNA, is known to be effective for the KO of *α-cat*, having been previously targeted in dogs and humans [21,22]. The oligonucleotides were inserted into the *Sap* I site of pCGSapI to create pCGSapI-*α-cat* (Figure 1B). The fidelity of the resultant pCGSapI-*α-cat* was confirmed by DNA sequencing using the primer upstream of the *Sap* I site.

To obtain *α-cat* KO P19 cells, P19 cells (5 × 10^8^) were co-transfected with pCGSapI-*α-cat* (~2 μg/μL) and pC-GFPHA (1.6 μg/μL) (Figure 1B), a vector that confers the expression of enhanced green fluorescent protein (EGFP) and HA tag, along with a neomycin resistance gene (*neo*), using the Ca-phosphate method as previously described [23]. After transfection and subsequent selection with G418, emerging colonies were isolated using a 200-μL pipette tip and propagated serially. The isolated clones were then checked for EGFP fluorescence under a fluorescence microscope and for α-cat expression by western blotting.

To obtain revertants of *α-cat* KO P19 cells, *α-cat* KO P19 cells (~10^8^) were electroporated with the pC-ha-cat vector (~2 μg/μL; Figure 1B; [24]), which confers the expression of normal HA-tagged *α-cat* under the *CAG* promoter, and the pC-bsr vector (~2 μg/μL; Figure 1B; which is the same of pCAG/bsr-7 [12,21]), which confers blasticidin S resistance under the *CAG* promoter, using the Lonza nucleofection system. After selection with blasticidin S (8 µg/mL), emerging colonies were isolated using the pipette tip method. The isolated clones were then checked for α-cat expression by western blotting.

### 2.3. Heat Map Analysis

Microarray analysis was performed on P19, C7, and Rev-C7 cells. Fluorescent labeling of RNA samples (total RNA) was performed using a 3D-Gene^®^ microRNA (miRNA) labeling kit (Toray Industries, Tokyo, Japan). The miRNA expression levels were measured using DNA chips (3D-Gene Mouse miRNA Oligo chip ver.21 [Mouse_miRNA_V21], Toray Industries). A heat map of miRNA expression levels was generated using microarray analysis. Blue, red, and yellow indicate low, high, and intermediate expression levels, respectively.

### 2.4. Generation of Embryoid Bodies (EB)s and Retinoic Acid (RA)-Based Induction for Cellular Differentiation

For EB formation, P19 cells and their derivatives were detached and dissociated into single cells with 0.25% trypsin-EDTA, and then plated onto a 35-mm non-coated dish (#1000-035; Iwaki Glass Co. Ltd., Tokyo, Japan) in 3 mL of DMEM-high glucose/10% FBS at a density of 5 × 10^4^ cells/mL to form aggregates. On day 2, various concentrations (10^−8^ M and 10^−5^ M) of all-trans-RA (#R2625; Sigma-Aldrich Co., St. Louis, MO, USA) were added to the culture medium. The RA was reconstituted in 100% ethanol to prepare a stock solution.

EBs were photographed at different times (d-4, d-6, and d-8 after culture), and some EBs were collected for immunostaining for anti-SSEA-1. In some cases, a few EBs were picked and placed onto the adhesive substrate in a 35-mm gelatin-coated tissue-culture dish (#4000-020; Iwaki Glass Co., Ltd., Tokyo, Japan) to allow them to spread out.

### 2.5. In Vivo Teratoma Formation Assay

Cell grafting into murine pancreatic tissue was initiated using a method (intrapancreatic parenchymal cell transplantation (IPPCT)) described by Sato et al. [25]. P19 cells and their derivatives were harvested by trypsinization and suspended in Dulbecco’s Ca^2+^ -Mg^2+^-free phosphate-buffered saline (D-PBS(−)), pH 7.4 (#14249–95; Nacalai Tesque, Inc., Tokyo, Japan). The cells were diluted to ~5 × 10^4^ cells/µL in 20 µL of D-PBS(−) and 0.1% (*v*/*v*) trypan blue (Trypan Blue Stain 0.4%; Invitrogen Co.; for visualization of the injected solution) in a 1.5-mL tube. The cell-containing solution (~2 µL) was aspirated using an injection micropipette connected to the mouthpiece under a dissecting microscope. The injection micropipette was then inserted into the internal part of the pancreatic tissue of the anesthetized nude mice under a dissecting microscope, and the cell-containing solution was injected. One to 1.5 months post-transplantation, emerging teratomas were harvested and fixed with 4% paraformaldehyde (PFA) in D-PBS(−) at 4 °C for 1 week. The fixed tissues were then dehydrated by immersion in 0.25% sucrose in D-PBS(−) at 4 °C for two days and then dehydrated in 0.4% sucrose in PBS at 4 °C for four days. These samples were then embedded in an optimum cutting temperature (O.C.T.) compound (Tissue-Tek^®^ [no. 4583]; Miles Scientific, Naperville, IL, USA) for cryostat sectioning. Cryostat sections were stained with hematoxylin and eosin (H&E). For the quantitative analysis of H&E-stained samples, the number of differentiated and undifferentiated cells within the defined box was counted manually and represented as a percentage of the total number of cells.

### 2.6. Formation of Aggregation Chimeras between EC Cells and an 8-Cell Embryo

Superovulated B6C3F1 females were mated with B6C3F1 males in the evening. The presence of vaginal plugs was checked the next morning, designated as Day 0.4 (10:00 AM). Eight-cell embryos were isolated by flushing oviducts from pregnant B6C3F1 females on Day 2.5. The isolated embryos underwent ZP removal via a short incubation in acid Tyrode’s solution [26]. To enhance embryo adhesiveness, the ZP-removed 8-cell embryos were briefly immersed in a solution (60 µL) containing phytohemagglutinin-P (PHA-P; Wako; #161-15251; 50 mg; final concentration 0.5 mg/mL) in D-PBS(−) + 0.3% FBS for approximately 0.5 min at room temperature. Subsequently, they were placed onto a substratum of a bacteriological dish with >400 trypsin-EDTA-dissociated cells (C7 or P19 cells) spread in a small drop (60 µL) (Appendix A Appendix A). PHA-P transiently enhances mouse 8-cell embryo adhesion to cells, as previously demonstrated for blastomere aggregation in mouse chimeras [27]. Several minutes after embryo attachment, each embryo was picked up using a breath control micropipette and transferred to a well of a Terasaki microtest plate (3–4 embryos per well). Each well contained 7 µL of modified Whitten’s medium covered with paraffin oil to prevent medium evaporation. Embryos with more than three cells on their surface were selected for further incubation in a 37 °C incubator with 5% CO_2_ in the air for 2 days. During this cultivation period, green fluorescence generated from embryonic stem cells (EC cells) was examined using a confocal laser scanning microscope (Carl Zeiss Japan, Tokyo, Japan). GFP fluorescence enabled assessment of pluripotency by integrating or excluding cells during chimera formation.

### 2.7. Western Blotting

Cell lysates were boiled in SDS sample buffer (0.0625 M Tris-HCl, pH 6.8, 2.3% SDS, 10% glycerol, and 5% 2-mercaptethanol), separated by SDS-PAGE, and transferred onto nitrocellulose membranes (Millipore, Billerica, MA, USA), as described by Ozawa [21]. Membranes were blocked with 5% skim milk in TBST (Tris-buffered saline + 0.1% (*v*/*v*) Tween 20) for 1 h and subsequently exposed to primary antibodies (rabbit polyclonal antibodies against α-cat (#C2081; Sigma-Aldrich; 1:100) and monoclonal antibody (mAb) against α-tubulin (#T9026; Sigma-Aldrich; 1:500)) for 2 h or overnight, and then to secondary antibodies (obtained from Jackson ImmunoResearch Laboratories, West Grove, PA, USA)) for 1 h. Proteins were detected using an ECL Plus system (GE Healthcare, Saint Louis, MO, USA). The reproducibility of the results was confirmed by repeated experiments and a representative blot is shown.

### 2.8. Immunostaining

Cells (EBs) were fixed with 4% PFA for 10–20 min at 4 °C, and then washed twice with PBS containing 1% normal goat serum (NGS) (Invitrogen) (hereafter referred to as D-PBS(−)/NGS). The specimens were then blocked with 20% AquaBlock tm/EIA/WB (#PP82; East Coast Biologics, Inc., North Berwick, ME, USA) for 30 min at 4 °C, prior to incubation with the primary antibody SSEA-1 (1:500) (#ab16285; Abcam, Cambridge, UK) for 24 h at 4 °C. Notably, the specificity of the anti-SSEA-1 antibody used in this study was verified by co-incubation with 1 mM fucose, which resulted in a complete loss of staining activity [28]. After washing twice with D-PBS(−)/NGS, the samples were incubated with secondary antibody Alexa Fluor 594-conjugated goat anti-mouse IgM (1:200) (#A21044; Invitrogen) for 3 h in a blocking buffer at 4 °C. After two washes with D-PBS(−)/NGS, the samples were subjected to fluorescence observation.

### 2.9. Fluorescence Observation

The fluorescence in the samples was examined under an Olympus BX60 fluorescence microscope using DM600 (BP545–580 and BA6101F; Olympus, Tokyo, Japan) filters, which were used to detect Alexa Fluor 594-derived red fluorescence. Micrographs were recorded using a digital camera (FUJIX HC-300/OL; Fuji Film, Tokyo, Japan) attached to a fluorescence microscope, and the images were printed using a Mitsubishi digital color printer (CP700DSA; Mitsubishi, Tokyo, Japan). In some cases, confocal laser scanning microscopy was used to detect green fluorescence in chimeric embryos.

### 2.10. Statistical Analysis

The differentiated cells generated at the edge of a colony were analyzed by ImageJ (ImageJ bundled with 64-bit Java 8; National Institutes of Health, Bethesda, MD, USA) to measure the cell area, with results represented as graphical image (box plot). Statistical analysis was performed using GraphPad PRISM 5 for Windows software version 11 (GraphPad Software, Inc., La Jolla, CA, USA). Data were analyzed by performing one-way repeated measures ANOVA, and the results are expressed as mean ± SD; the statistical significance was defined as *p* < 0.05.

## 3. Results

### 3.1. Generation of α-Cat KO P19 Cell Lines and Their Revertant

P19 cells were first labeled with an EGFP transgene using pC-GFPHA, which carries an expression unit for EGFP cDNA linked to an HA tag at its 5′ end, along with a neomycin resistance gene (*neo*). The transfection was performed using a Ca-phosphate-mediated system. After selection with G418, the resulting fluorescent colonies were picked with a 200-μL pipette tip and propagated. To knock out the expression of endogenous mouse *α-cat* in P19 cells, we created an expression vector named pCGSapI-*α-cat*, which includes two expression units: one for the gRNA targeted against mouse *α-cat* and another for the *Cas9* gene. P19 cells were transfected with a solution containing pCGSapI-*α-cat* and pC-GFPHA using the Ca-phosphate method. After selection with G418, the resulting colonies were screened for EGFP expression, and at least two lines, designated C7 and C9, were successfully propagated. Both cell lines exhibited a round morphology with a loss of cell–cell contact when maintained on an adhesive substrate under either densely or sparsely seeded conditions (Figure 2A(b,c,e,f)). In contrast, the parental P19 cells displayed clear cell–cell contact under both conditions (Figure 2A(a,d)). Western blot analysis revealed that the parental P19 cells exhibited a distinct band for α-cat protein, whereas C7 did not (left column of Figure 2B), indicating a complete ablation of α-cat protein expression in C7. This result suggested that confirmation by genomic DNA sequencing for the presence of indels (insertion or deletion of nucleic acids) in C7 was unnecessary. Among these two lines, C7 exhibited better growth than C9, so the properties of C7 were examined in detail.

To test whether the loss of cell–cell contact observed in C7 was due to the lack of α-cat protein expression, C7 was transfected with an α-cat expression vector to obtain a “revertant of C-7 (rev-C7).” Specifically, C7 was transfected with a solution containing pC-bsr, a vector conferring expression of the blasticidin S resistance gene (bsr), and pC-ha-cat, a vector conferring expression of human α-cat, using an electroporation-mediated transfection system (Lonza). After selection with blasticidin S, the resulting colonies were propagated. One clone, when tested for its ability to show autonomous aggregation two days after cultivation under adhesive conditions, exhibited tight cell–cell contact similar to the parental P19 cells (Figure 2C(b) vs. Figure 2C(a–c). This suggested that rev-C7 had recovered α-cat expression. Western blot analysis of cell lysate from rev-C7 confirmed that rev-C7 indeed expressed α-cat protein (right column of Figure 2B). These data demonstrate that the reduced cell–cell adhesion activity in C7 was solely due to the ablation of α-cat. Additionally, miRNA microarray experiments were performed using total RNAs isolated from P19, C7, and rev-C7. The results showed no gross alterations in the gene expression patterns among these cells (Figure 2D). As rev-C7 cells are believed to resemble P19 cells in morphology and behavior, C7 cells were used for further studies, including susceptibility to cellular differentiation in response to RA, in vivo differentiation ability, and the potential to colonize early embryos.

### 3.2. EBs Generated after Culture on Non-Adhesive Dishes and Their Immunohistochemical Property

P19 cells cultivated in DMEM-high glucose/10% FBS on a non-coated dish for more than a day aggregate and form 3-D cell aggregates known as EBs. After seven days in culture, EBs typically develop a thin outer layer composed of differentiated cells surrounding an inner core [18]. When intact P19 cells were cultured under suspension conditions, they formed aggregates with tight cell–cell contacts (Figure 3A(a–c)). However, contrary to expectations, the outer surface layer remained undifferentiated, appearing smooth (Figure 3A(c)). Immunostaining for anti-SSEA-1, a monoclonal antibody specific to carbohydrate antigens expressed in ES/EC cells and early embryos [29], further confirmed the undifferentiated state of the outer surface and inner core of P19 EBs (arrows and asterisks in Figure 3B(a,b)). These results indicate that the P19 cells used in this study did not respond to differentiation cues from the surrounding environment, contrary to previous reports [18].

Next, we attempted to cultivate C7 cells under similar suspension conditions as used for P19 cells. While C7 cells formed aggregates, the surface of these aggregates was irregular, unlike that observed with P19 cells (Figure 3A(d–f)). Additionally, these structures were fragile, as vigorous pipetting easily and rapidly dissociated them into single cells. Immunostaining of C7 EBs cultured for 8 days with anti-SSEA-1 showed positive staining in both outer and internal cells (arrows and asterisks in Figure 3B(c,d)).

### 3.3. Forced Induction for Differentiation of EBs by Cultivating in RA-Containing Medium

The vitamin A derivative RA is known to be a potent inducer for the differentiation of EC cells, including P19 and F9, a nullipotent EC cell line [16,17,18,30]. In this case, the concentration of RA used in the treatment is an important factor in determining the type of differentiated cells formed from RA exposed EB [19,30]. For instance, Grover et al. [30] demonstrated that treatment of F9 cells with lower concentrations of RA (below 10^−8^ M) led to the generation of EBs with an outer layer of visceral endoderm cells that synthesized and secreted alpha-fetoprotein, whereas treatment with higher concentrations of RA (~10^−5^ M) resulted in the generation of mesodermal cells at the outer surface of an EB.

When P19 EBs were incubated with RA-containing medium for 8 days, differentiated cells with irregular shape were first discernible on the outer surface of the EBs (at d-4) in both groups (treatment with 10^−8^ M and 10^−5^ M RA) (Figure 4A(a,b)). These differentiated cells were also seen on the outer surface of the EBs (at d-6 and -8) in both groups (Figure 4A(e,f,i,j)). Immunostaining of P19 EBs cultured for 8 days with anti-SSEA-1 showed negative staining in the outer cells and inner cells in both groups (Figure 4B(a,b,e,f)). Cultivation of C7 EBs resulted in generation of irregular cells on the outer surface of the EBs (at d-4 to d-8) in the group of treatment with 10^−8^ M RA (Figure 4A(c,g,k)). Notably, the surface of C7 EBs (d-4 to d-8) appears indistinguishable from that of P19 EBs (d-4 to d-8), suggesting that the outer cells on C7 EBs during this period may comprise differentiated cells. However, immunostaining of C7 EBs cultured for 8 days with anti-SSEA-1 showed positive staining in the outer cells (arrows in Figure 4B(d)). Treatment with 10^−5^ M RA was deleterious to the growth of C7 cells (Figure 4A(d,h)). These results suggest that C7 is still in an undifferentiated state even after cultivation under conditions of strong differentiation induction. In other words, C7 is refractory to the differentiation-induction force.

These findings were further validated by observing cells spreading from the EBs after their attachment to an adhesive substrate, confirming that the cells at the colony edge were derived from cells localized at the outer surface layer of the EBs. Outgrowth of P19 EBs four days after attachment of EBs (grown for eight days in the presence of 10^−8^ M RA or 10^−5^ M RA) allowed us to generate flat or epithelial-like cells that did not resemble the parental P19 cells (Figure 4C(a’,b’)). In contrast, C7 EBs (treated with 10^−8^ M RA) generated round cells, similar to C7 cells (Figure 4C(c’)). Quantitative analysis of the cells at the edge of the colony confirmed that the morphology of differentiated cells in each group was significantly different from each other (Figure 4D).

### 3.4. In Vivo Teratoma Formation Assay through Transplantation of P19 and Its Derivative into Pancreatic Tissues

To test the multiple differentiation capabilities of pluripotent cells, transplantation of these cells into appropriate sites (as exemplified by subcutaneous grafting or transplantation underneath the kidney capsule) of immunocompromised mice (i.e., nude or SCID mice) has been frequently employed [31]. However, the traditional methods often lack reproducibility. To avoid this uncertainty, we developed a new method (IPPCT) for in vivo teratoma formation assay by grafting pluripotent cells onto the pancreatic parenchyma of nude mice under a dissecting microscope using a mouthpiece-controlled micropipette (schematically shown in Figure 5A). This approach requires only a small number of grafted cells (less than 100), and the efficiency of tumor formation is 100% [25].

Using this technique, we grafted C7 or P19 cells (which were maintained in a normal medium (DMEM-high glucose/10% FBS), into nude mice. One and a half months after IPPCT, solid tumors with diameters of approximately 1 cm were obtained from both grafts. The tumors were subjected to cryostat sectioning. In the H&E-stained samples derived from the grafting of P19 cells, some differentiated cells appeared as clusters and were clonally expanded (enclosed by dotted lines in Figure 5B(a). Figure 5B(a) presents at least four clusters: enlarged cells with vacuolated cytoplasm (18%; 36/199; #1), enlarged cells with translucent cytoplasm (39%; 78/199; #2), fibroblastic cells (18%; 36/199 tested; #3), and EC-like cells similar to P19 cells (25%; 49/199; #4). In contrast, no distinct differentiated cells were detected in the sample from C7 cell graft (Figure 5B(b)), where all cells appeared EC-like similar to P19 cells (314/314).

### 3.5. EC Cells/8-Cell Embryo Aggregation Assay

Since pluripotent cells such as ES and EC cells are similar to early embryos, they tend to colonize into the early embryos when they closely attached to the surface of 8-cell embryos or an inner cell mass (ICM) facing the blastocoel of a blastocyst [32,33,34,35]. To test whether C7 can colonize early embryos, an aggregation assay between EC cells and 8-cell embryos was performed. First, ZP-removed 8-cell embryos were treated with PHA-P, a glycoprotein that binds to cellular surfaces by means of specific glycol conjugates [36], for a short period and then placed onto > 400 trypsin-EDTA dissociated cells (C7 or P19) that had been spread onto a substratum of the bacteriological dish in a small drop (60 μL) (Appendix A). Several minutes after attachment of an embryo to the cells, each embryo was picked up using a breath-control micropipette and transferred to a well of a Terasaki microtest plate (3–4 embryos/well) containing egg culture medium. The plate was then incubated in an incubator with 5% CO_2_ in air at 37 °C for 3 days. During this cultivation period, the green fluorescence generated by the EC cells was observed using a fluorescence microscope. EGFP fluorescence enables assessment of the behavior of EC that have been integrated or excluded during chimera formation.

On the 1st day after aggregation, almost all the morulae generated after aggregation with intact P19 cells had one or more P19 cells on their surfaces or internal areas (arrows in Figure 6A(e–h,e’–h’)). When the number of morulae with one or more intact P19 cells was calculated and plotted on the graph, their percentage reached 92% (11/12 tested) (“P19/8” in Figure 6B). In contrast, almost all C7 failed to adhere to the surface of the ZP-removed embryos (arrowheads in Figure 6A(b–d,b’–d’)), and only a few embryos were found to have cells attached to the embryo surface (arrow in Figure 6A(a,a’)), but there were no embryos with it in its internal area (Figure 6A(a–d,a’–d’)). The percentage of embryos with one or more intact P19 cells was below 20% (2/11 tested) (“C7/8” in Figure 6B). On the 2nd day after aggregation, there was no embryo retaining C7 in its surface or internal area (Figure 6A(i–l,i’–l’); “C7/8” in Figure 6B), while all embryos retained one or more P19 cells in their surface or inside an embryo (Figure 6A(m–p,m’–p’); “P19/8” in Figure 6B).

## 4. Discussion

α-cat is a crucial component of the cadherin−catenin cell−cell adhesion complex that facilitates cell−cell contact in epithelial cells. Normally, this complex is primarily localized at plasma membrane sites involved in lateral cell−cell contacts, such as AJs. Depletion of α-cat expression leads to diminished cell−cell adhesion, causing the cadherin−catenin complex to redistribute from plasma membrane contact sites to a uniform distribution within the cell [12]. This disruption of AJs eliminates “contact inhibition of cell proliferation”, allowing cells to proliferate autonomously, potentially giving rise to tumorigenic cells [37,38]. Loss of cell−cell contact may also promote the metastasis of tumorigenic cells [37].

The significance of cell–cell contact in organ morphogenesis and embryogenesis has been extensively studied [39,40,41]. For example, ectopic overexpression of the cytoplasmic domain of E-cadherin in C2C12 myoblasts disrupts cell–cell contact, causing a morphological shift from spindle-like to rounded, thereby inhibiting cell–cell fusion critical for myotube formation [12]. Conditional KO experiments using trap vector knock-in strains have revealed various outcomes: (1) germ cell loss upon gonad-specific *Cdh1 KO* [42], (2) developmental arrest at the blastocyst stage due to *α-cat* or *Cdh1* homozygous KO in trophectodermal cells [7,8,11], (3) progressive hyperplasia in the skin epithelium upon *Cdh1*-specific KO [43], (4) loss of polarity and hyperproliferation in skin epithelium upon *α-cat*-specific KO [44], and (5) severe disruption of intestinal morphogenesis upon *Cdh1*-specific KO in intestinal epithelium [45].

In this study, we demonstrated that loss of cell–cell contact inhibits initial cellular differentiation in *α-cat* KO P19 cell clones (C7) during RA-induced differentiation, as evidenced by persistent SSEA-1 reactivity in RA-treated C7 cells (Figure 4B(c,d)). Histological analysis of solid tumors derived from grafts of C7 cells into nude mice revealed predominantly undifferentiated cells, contrasting with P19 cell-derived tumors that exhibited both differentiated and undifferentiated cells (Figure 5B(b) vs. Figure 5B(a)). These findings suggest that cell–cell contact is crucial for early stages of cellular differentiation, such as the formation of primitive endoderm.

Although alteration in expression of other genes related to the cadherin-catenin adhesion system could potentially contribute to loss of cell–cell contact in *α-cat* KO P19 cells, miRNA microarray experiments (heat map analysis) showed no overt changes in gene expression among P19, C7, and revertant C7 (rev-C7) cells (Figure 2D). Introduction of normal *α-cat* gene into C7 cells restored normal cell–cell contact (Figure 2C(c)). Additionally, studies by Ozawa et al. [21] demonstrated that *α-cat* KO canine cells (MDCK clone) exhibited no significant differences in E-cadherin, β-catenin, and vinculin expression levels compared to wild-type cells by western blotting. Similarly, *α-cat* KO blastocysts showed unaltered expression and distribution of E-cadherin and β-catenin in trophectodermal cells [11]. These findings suggest that loss of cell–cell contact in *α-cat* KO P19 cells is primarily due to *α-cat* deficiency.

Previous studies have highlighted the pluripotent nature of cells such as ES/EC/induced pluripotent stem (iPS) cells and embryonic cells (ICM and epiblasts), characterized by their ability to contribute to the development of embryos from 8-cell stage to blastocysts [46]. However, recent experiments have demonstrated that multipotent fetal and adult stem cells (including bone marrow and neuronal stem cells) and somatic mouse embryonic fibroblasts can also participate in chimera formation [46]. Burnside and Collas [47] observed that while ES and HC-11 epithelial cells formed gap junctions with blastomeres within 24 h of culture, NIH3T3 fibroblasts required reprogramming with HC-11 cell extracts to achieve similar integration, owing to their lack of E-cadherin expression [48]. Introduction of *E-cadherin* gene into NIH3T3 fibroblasts may facilitate their incorporation into embryos via gap junction formation with blastomeres.

We used P19 cells in this study to investigate the role of cell-cell contact in cellular differentiation due to their multipotent nature [14]. However, the differentiation potential of the P19 cells used here appeared limited, as evidenced by the failure of P19 EBs to generate SSEA-1-negative cells in the outer layer after suspension culture (arrows in Figure 3B(a,b)), and the presence of poorly differentiated cells in solid tumors generated by in vivo grafting of P19 cells (Figure 5B(a)). Further studies using iPS or ES cells, which exhibit greater propensity for multilineage differentiation, are warranted to validate these findings.

## 5. Conclusions

Cell–cell contact mediated by the cadherin–catenin cell–cell adhesion complex plays a critical role in early mammalian embryogenesis. Disruption of this system, such as through KO of *E-cadherin* or *α-cat*, results in morphological changes from polygonal/epithelial to round-shaped morphology and leads to unregulated cellular proliferation. However, it has remained unclear whether disruption of the cadherin–catenin cell–cell adhesion complex inhibits cellular differentiation during earlier stages of development. To address this, we utilized the CRISPR/Cas9 system to block α-cat expression in P19 cells, a pluripotent cell line that mimics early embryogenesis. The resulting *α-cat* KO P19 cells showed an inability to differentiate even under conditions conducive to potent differentiation induction. These findings lead us to conclude that the cadherin–catenin cell–cell adhesion complex is indispensable for cellular differentiation during the early stages of embryonic development.

## Figures and Tables

**Figure 1 biotech-13-00041-f001:**
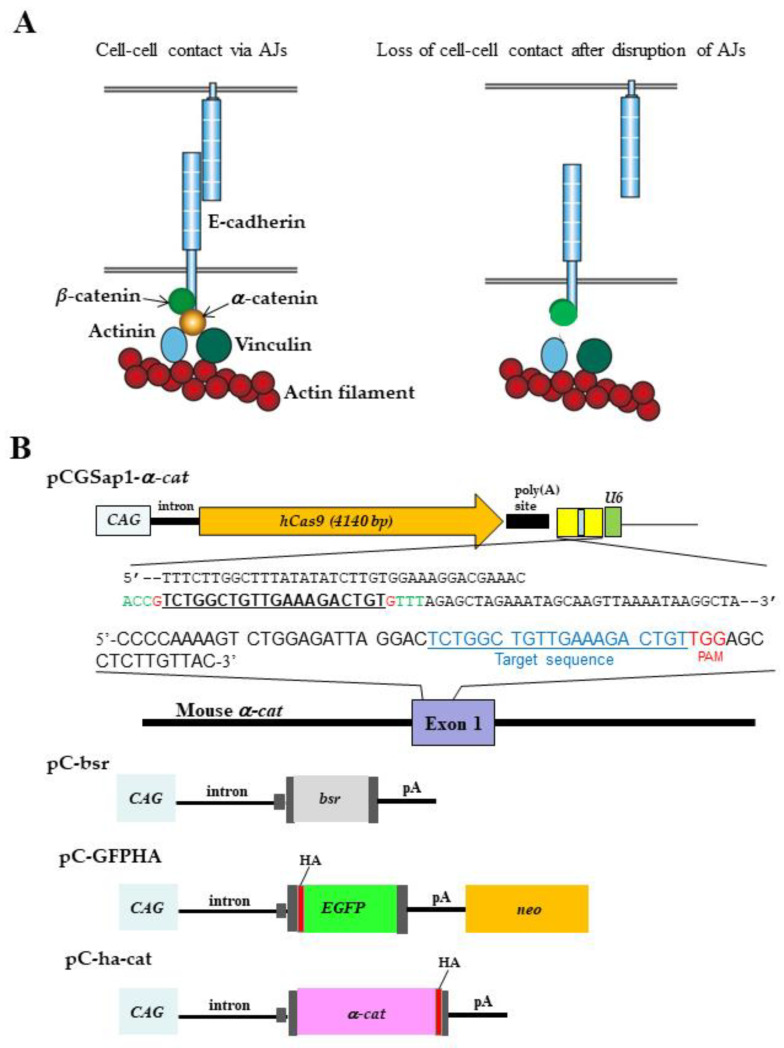
Molecules required for cell–cell contact and plasmids for the generation of genetically engineered P19 cells. (**A**): Multiprotein complexes at adherens junctions (AJs). α-Catenin (α) is one of the essential components in the E-cadherin–catenin cell adhesion complex, which mediates cell–cell contact via the formation of AJs. It forms a bridge between the E-cadherin and cytoskeletal actin filaments. Once α is depleted from a cell, the bridge would be disrupted together with loss of mechanotension, leading to loss of cell–cell contact. β, β-catenin. (**B**): The vectors for yielding *α-cat* knock out (KO) P19 cells using the CRISPR/Cas9 system or those for rescuing *α-cat* KO cells. Structure of pCGSap1 carrying a *Cas9* expression unit (comprising *CAG*, the second intron of the rabbit *β-globin* gene, the humanized *Cas9* (*hCas9*) gene, and the poly(A) site of the rabbit *β-globin* gene) and a guide RNA (gRNA) expression unit (comprising *U6*, multiple sites into which chemically synthesized gRNA is inserted, and a poly(A) site). Oligonucleotides targeting *α-cat* exon 1 were synthesized and inserted into the *Sap* I site of pCGSap1 to create pCGSap1-*α-cat*. The underlined sequence corresponds to the sequence of gRNA targeting the *α-cat* gene. Nucleotides (shown in red and green) at both the 5′ and 3′ ends of the gRNA are those recognized by *Sap* I. pC-bsr is a vector conferring expression of the blasticidin S resistance gene (*bsr*). pC-GFPHA is a vector that confers the expression of enhanced green fluorescent protein (EGFP) and an HA tag (HA). pC-ha-cat is a vector conferring expression of human α-cat cDNA and HA. CAG, chicken β-actin based promoter; *U6*, human *U6* promoter. *neo*, neomycin resistance gene expression unit; HA, HA tag; pA, poly(A) site.

**Figure 2 biotech-13-00041-f002:**
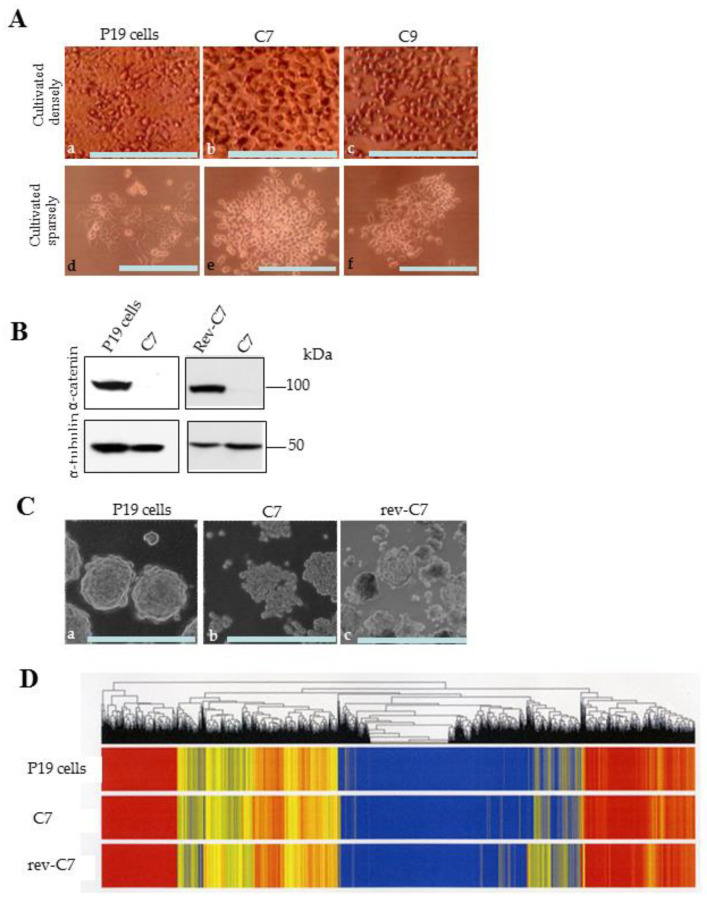
Morphology of the genetically engineered P19 cells, western blotting, and RNA expression profiling. (**A**): Cell behavior of P19 cells (a,d) and *α-catenin* (*α-cat*) KO P19 cells [C7 (b,e) and C9 (c,f)] 2 days after seeding onto an adhesive substrate densely (a–c) or sparsely (d–f). Bar: 100 μm. (**B**): Immunoblot detection of α-cat in P19, C7, and revertant (rev-C7) cells. α-Tubulin was used as a loading control. (**C**): Cell behavior of P19 cells (a) and *α-cat* KO P19 cells (C7) (b) and revertant (rev-C7) (c) 4 days after seeding onto an adhesive substrate densely. Bar: 100 μm. (**D**): Heatmap analysis of P19, C7, and rev-C7 cells.

**Figure 3 biotech-13-00041-f003:**
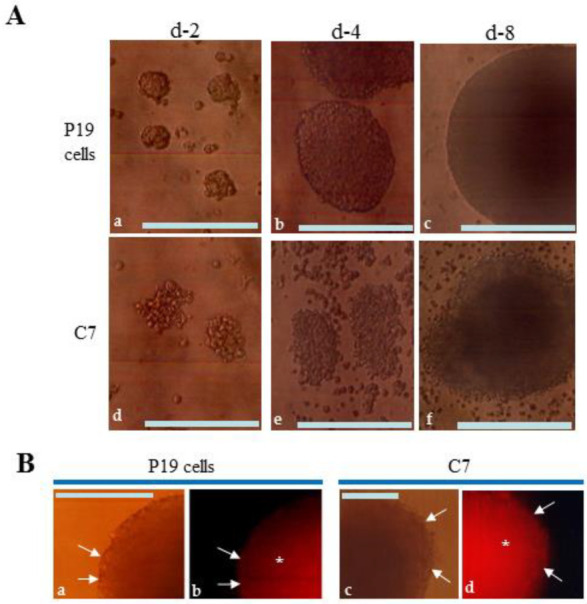
Embryoid body (EB) formation of P19 cells. (**A**): EB formation by simple suspension culture of P19 cells (a–c) and C7 cells (d–f) in a non-coated dish for 2 (a,d), 4 (b,e), and 8 d (c,f). Over 10 EBs were checked and they all showed similar morphology. Bar: 100 μm. (**B**): Immunostaining with an anti-SSEA-1 antibody. EBs were generated from P19 cells (a,b) or C7 cells (c,d) after cultivation in DMEM-high glucose/10% FBS in a non-adhesive dish for 8 days. Arrows and asterisks indicate positive staining with anti-SSEA-1. (a,c): Photographs captured under light; (b,d): Photographs captured under UV illumination. Bar: 100 μm.

**Figure 4 biotech-13-00041-f004:**
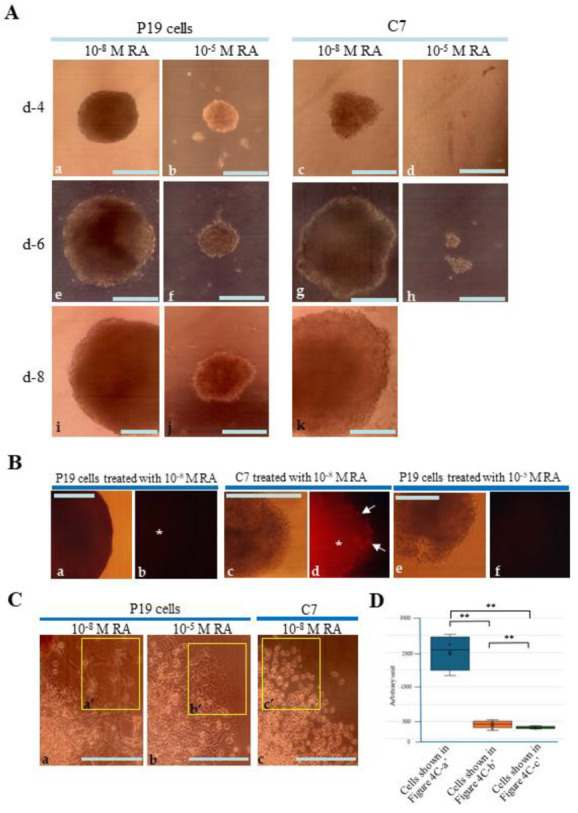
Retinoic acid (RA)-induced differentiation of P19 embryoid bodies (EBs). (**A**): EBs generated from P19 cells or C7 after cultivation in a non-adhesive dish in the presence of RA at various concentrations (10^−8^ or 10^−5^ M) for 4 days (a–d), 6 days (e–h), and 9 days (i–k). Bar: 100 μm. (**B**): Immunostaining with an anti-SSEA-1 antibody. EBs generated from P19 cells (a,b,e,f) or C7 cells (c,d) after cultivation in the presence of 10^−8^ M (a–d) or 10^−5^ M RA (e,f) in a non-adhesive dish for 8 days. The outer layer surrounding the core cells (indicated by asterisks) of P19 EBs treated with 10^−8^ M or 10^−5^ M RA was completely negative for anti-SSEA-1 staining (a,b,e,f). In contrast, C7 EBs treated with 10^−8^ M RA were reactive to anti-SSEA-1 in both the outer layer (arrows in d) and the inner core (asterisks in d). Bar: 100 μm. (**C**): Outgrowth of P19 cells (a,b) and C7 cells (c) four days after attachment of EBs [grown for eight days in the presence of 10^−8^ M (a,c) or 10^−5^ M RA (b)] to the adhesive surface. Various types of differentiated cells [elongated cells (a’) vs. flat cells (b’)] were detected when P19 cells were cultured with different concentrations of RA. In contrast, no differentiated cells (c’) were detected when C7 cells were cultivated with 10^−8^ M RA. Bar: 100 μm. (**D**): Quantitative analysis of cells at the edge of a colony. Cells in the boxes (a’–c’) presented in Figure 4C were subjected to image analysis and plotted. Over 6 (for P19 cells treated with 10^−8^ M RA) and 10 cells (for P19 cells treated with 10^−5^ M RA and C7 treated with 10^−8^ M RA) were selected randomly and plotted. **: *p* < 0.05.

**Figure 5 biotech-13-00041-f005:**
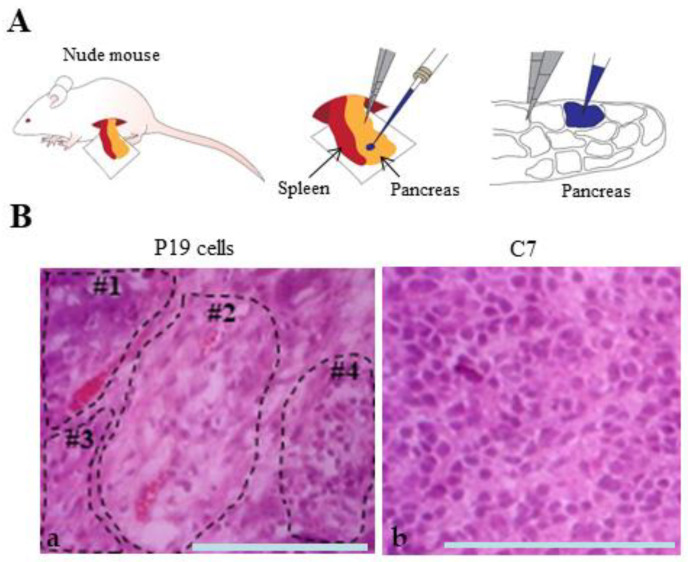
In vivo teratoma formation. (**A**): Schematic illustration of in vivo teratoma formation by inoculation of tumor cells into the pancreatic parenchyma of nude mice. First, the spleen and pancreas were removed from anesthetized mice. Then, a solution (1~2 μL) containing tumor cells and trypan blue (used as visible marker) is injected into the inner area of pancreas using a breath-controlled micropipette under a dissecting microscope. This procedure is developed by Sato et al. [25] and named “IPPCT”. (**B**): Hematoxylin-eosin (H&E)-stained cryostat sections of solid tumors generated 1.5 months after IPPCT with P19 cells (a) or C7 cells (b). Differentiated cells are indicated by dotted lines (numbered #1–#3) together with undifferentiated cells (numbered #4) in the P19 cell-derived tumors (a). In contrast, no distinctly differentiated cells were visible in the C7-derived tumors (b). Bar: 100 μm.

**Figure 6 biotech-13-00041-f006:**
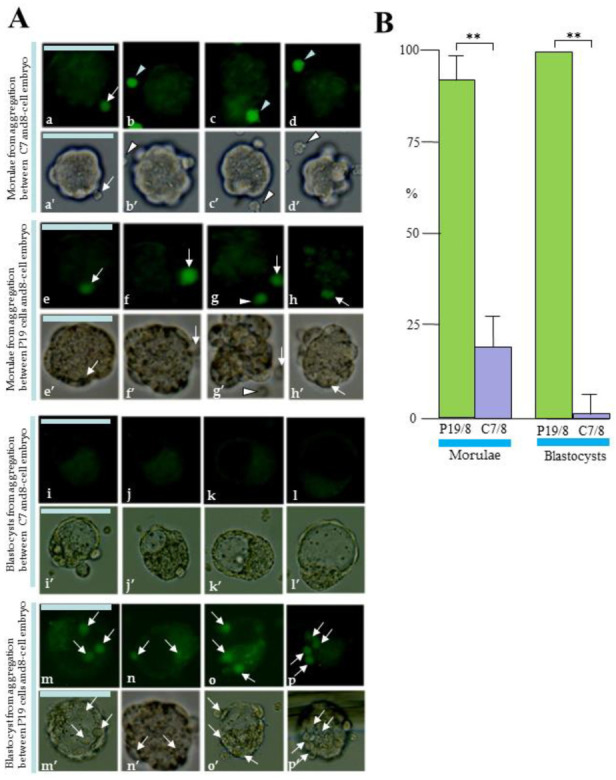
Formation of aggregation chimeras between embryonal carcinoma (EC) cells and 8-cell embryos. (**A**): Location of EC after aggregation of chimeras between P19 cells ((e–h), (e’–h’), (m–p), (m’–p’)) or C7 ((a–d), (a’–d’), (i–l), (i’–l’)) and 8-cell embryos. Since EC cells were labelled with EGFP, which is useful for tracking chimeras, green fluorescence in EC cells was checked at the morula ((a–d), (a’–d’), (e–h), (e’–h’)) and blastocyst ((i–l), (i’–l’), (m–p), (m’–p’)) stages after EC cell/embryo aggregation. Arrows indicate EC attached to the surface of an embryo or incorporated into its internal area. Arrowheads indicate that EC are present near the embryo but are not attached to its surface. (a–d), (e–h), (i–l), (m–p): photographed under UV illumination; (a’–d’), (e’–h’), (i’–l’), (m’–p’): photographed under light. Bar: 100 μm. (**B**): Graphs showing the percentage of embryos with successful aggregation between EC cells and an embryo per total number of embryos tested. Successful aggregation was determined when embryos with one or more EC on the surface or internal area of the embryo were recognized. P19/8, chimera between P19 cells and 8-cell embryo; C7/8, chimera between C7 and 8-cell embryo. **: *p* < 0.05.

## Data Availability

The original contributions presented in the study are included in the article; further inquiries can be directed to the corresponding author.

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
