# Peer review of "Loss of Cell-Cell Contact Inhibits Cellular Differentiation of α-Catenin Knock Out P19 Embryonal Carcinoma Cells and Their Colonization into the Developing Mouse Embryos"

_biotech, 2024, doi:10.3390/biotech13040041_

Round 1

Reviewer 1 Report

Comments and Suggestions for Authors

Thank you for submitting your manuscript to bio Tech journal 

The idea is exciting although discussed before as in Terry Lechler,2019

using the IPPCT assay the author developed before to induce teratoma formation was novel.

figure 1A:  alpha, and beta-catenin need to be assigned correctly instead of just alpha and beta

Figure 7b error bars need to be added 

Comments on the Quality of English Language

minor typos need to be corrected

Author Response

Reviewer-1: Comments and Suggestions for Authors

Thank you for checking our manuscript. In the revised text, the immune-stained data shown in Figure 6 were combined to Figures 3 and 4 to understand the revised text more clearly. As a result, the total number of figures was reduced from 7 to 6 in the revised text.  

Thank you for reviewing our manuscript and providing insightful comments, which have helped us improve our manuscript significantly. In the revised manuscript, we have combined the data of immunostaning, previously presented in Figure 5, in Figures 3 and 4 to improve clarity and coherence. The revised manuscript now includes six figures.

Thank you for submitting your manuscript to bioTech journal. The idea is exciting although discussed before as in Terry Lechler, 2019 using the IPPCT assay the author developed before to induce teratoma formation was novel.

Question-1: figure 1A:  alpha, and beta-catenin need to be assigned correctly instead of just alpha and beta.

Answer-1: As suggested, alpha- and beta-catenin have been included in Figure 1A of the revised manuscript.

Question-2: Figure 7b error bars need to be added.

Answer-2: Thank you for the suggestion. Accordingly, we have added error bars to Figure 6B (formerly Figure 7b) in the revised manuscript.

Question-3: Comments on the Quality of English Language

minor typos need to be corrected

Answer-3: The language of the manuscript was partly edited by an external language editing service providing company (Editage). We have corrected existing typographical errors in the revised manuscript.

Reviewer 2 Report

Comments and Suggestions for Authors

This manuscript from an established investigator, Professor Ozawa, why is a respected leader in the field of epithelial cell-0cell adhesions, addresses an important question regarding the effects of cell-cell adhesion in cell differentiation. The authors selected a bold way to disrupt intercellular junctions by knocking out a key adherens junction protein, alpha-catenin (aCAT) in P19 murine embryonic carcinoma cells. They report disruption of cell-cell contacts in aCAT-deficient cells and inability of these cells to undergo a stimulated differentiation in vivo and in vitro. Overall the study is descriptive but it presents interesting results and could be important for the future mechanistic studies. I still have a few relatively minor suggestions that could improve the data presentation.

Comments:

1. The presented results need quantification. The present manuscriot contains collection of pictures showing different structures formed by aCAT and control cells. The authors need to put some numbers for these pictures. For example, cell outgrowth at the colony edges could be quantified in Figure 3. Furthermore, RA-induced cell shape change could be quantified in Figure 4B. Finally, number of differentiated cells should be counted in H&E images presented in Figure 6.

2. Data presented in Figure 4 are not convincing. I do not see significant differences in the effects of RA on cell aggregates formed by aCAT-null and control cells. The authors should better emphasize such differences.

3. Immunolabeling data presented in Figure 5 is poor quality and need to be improved. It is unclear whether presented labeling of SSEA-1 is specific or this is just background. Better images and appropriate negative control are warranted.

4. Please describe the source of P19 cells.

5. Statistical analysis is missing for the data presented in Figure 7B graphs.

Author Response

Reviewer-2: Comments and Suggestions for Authors

Thank you for reviewing our manuscript and providing insightful comments, which have helped us improve our manuscript significantly. In the revised manuscript, we have combined the data of immunostaning, previously presented in Figure 5, in Figures 3 and 4 to improve clarity and coherence. The revised manuscript now includes six figures.

This manuscript from an established investigator, Professor Ozawa, why is a respected leader in the field of epithelial cell-0cell adhesions, addresses an important question regarding the effects of cell-cell adhesion in cell differentiation. The authors selected a bold way to disrupt intercellular junctions by knocking out a key adherens junction protein, alpha-catenin (aCAT) in P19 murine embryonic carcinoma cells. They report disruption of cell-cell contacts in aCAT-deficient cells and inability of these cells to undergo a stimulated differentiation in vivo and in vitro. Overall the study is descriptive but it presents interesting results and could be important for the future mechanistic studies. I still have a few relatively minor suggestions that could improve the data presentation.

Comments:

Question-1: The presented results need quantification. The present manuscript contains collection of pictures showing different structures formed by aCAT and control cells. The authors need to put some numbers for these pictures.

Answer-1: As suggested, the number of EBs analyzed has been described in the revised text (L344).

Question-2: For example, cell outgrowth at the colony edges could be quantified in Figure 3.

Answer-2: The results presented in the previous Figure 3B have been withdrawn because the cells that had outgrown from normal P19 cells were difficult to distinguish from the mutated P19 cells (C7). 

Question-3: Furthermore, RA-induced cell shape change could be quantified in Figure 4B.

Answer-3: As suggested, RA-induced cells at the colony edge were quantified by measuring the cell area using ImageJ software. The results are presented as a graph in Figure 4D of the revised text (L235-242; 364-367; 389-391).

Question-4: Finally, number of differentiated cells should be counted in H&E images presented in Figure 6. 

Answer-4: As suggested, the number of differentiated cells in the H&E-stained sections presented in Figure 5 (corresponding to the previous Figure 6) have been quantified and the results been added in the revised manuscript (L178-181; 408-414).

Question-5: Data presented in Figure 4 are not convincing. I do not see significant differences in the effects of RA on cell aggregates formed by aCAT-null and control cells. The authors should better emphasize such differences.

Answer-5: I agree with the reviewer’s comment. It was challenging to distinguish the outer surface of P19 EBs from that of C7 EBs after RA treatment. However, immunostaining with anti-SSEA-1, which specifically recognizes undifferentiated cells, revealed that the outer surface of P19 EBs (but not of C7 EBs) failed to react with anti-SSEA-1. This suggests the presence of differentiated cells in RA-treated P19 EBs. These observations have been mentioned in the revised manuscript (L375-378 and L378-379).

Question-6: Immunolabeling data presented in Figure 5 is poor quality and need to be improved.

Answer-6: The previous images have been replaced with those showing clearer staining for anti-SSEA-1 (please see Figures 3B and 4B in the revised manuscript).  

Question-7: It is unclear whether presented labeling of SSEA-1 is specific or this is just background. Better images and appropriate negative control are warranted.

Answer-7: To confirm the specificity of anti-SSEA-1, we employed a co-incubation approach with anti-SSEA-1 and high concentrations of fucose, which is recognized by the anti-SSEA-1 antibody. After co-incubation, we confirmed the loss of anti-SSEA-1 immunostaining activity, demonstrating antibody specificity. These details have been included in the revised manuscript (L221-223).

Question-8: Please describe the source of P19 cells.

Answer-8: As suggested, the source of the P19 cells has now been provided in the revised manuscript (L107).

Question-9: Statistical analysis is missing for the data presented in Figure 7B graphs.

Answer-9: As suggested, the data shown in Figure 7B was statistically analyzed. The results, including error bars and p value, have been described in Figure 6B (formerly Figure 7B) of the revised manuscript.

Reviewer 3 Report

Comments and Suggestions for Authors

In this manuscript, the authors used CRISPR/Cas9 to generate a-catenin knock out P19 embryonal carcinoma cells to demonstrate the effect of cadherin-catenin cell-cell adhesion in early stages of cellular differentiation. Although this study is interesting, several issues remain to be addressed:

-          As a-catenin KO clones were acquired, did the author performed off-target screening?

-          The author showed cell morphology, WB and RNNA-seq of Rev-C7 in Fig. 2, but why not include the data of this cell line in the following figures?

-          In Fig. 4A, the EB size of P19 and C7 (KO) in d6 and d8 upon 10^-8M RA administration didn’t show big difference while 10^-5M RA on C7 showed high toxic. Did the author compare the effect of P19 and C7 in 10^-7 and 10^-6 M RA treatment? Please clarify this.

-          Add the molecular weight in Figure 2B.

Minor errors:

-          Fig. 2A: amend C8 to C9 according to line 243 on page 4, amend image ‘a,c’ to ‘a,d’.

Author Response

Reviewer-3: Comments and Suggestions for Authors

Thank you for reviewing our manuscript and providing insightful comments, which have helped us improve our manuscript significantly. In the revised manuscript, we have combined the data of immunostaning, previously presented in Figure 5, in Figures 3 and 4 to improve clarity and coherence. The revised manuscript now includes six figures.

In this manuscript, the authors used CRISPR/Cas9 to generate a-catenin knock out P19 embryonal carcinoma cells to demonstrate the effect of cadherin-catenin cell-cell adhesion in early stages of cellular differentiation. Although this study is interesting, several issues remain to be addressed:

Question-1: As a-catenin KO clones were acquired, did the author performed off-target screening?

Answer-1: We did not perform off-target screening to identify potential sites affected by the KO of α-catenin gene in the isolated clones. However, we obtained a revertant, termed as rev-C7 cells, by introducing normal human α-catenin cDNA into the genome of C7 cells. These rev-C7 cells resembled the parental cells P17 in morphology, cell behavior, and gene expression profile. Therefore, we speculate that there were no detectable side effects on the properties of the P19 cells after genome editing.    

Question-2: The author showed cell morphology, WB and RNNA-seq of Rev-C7 in Fig. 2, but why not include the data of this cell line in the following figures?

Answer-2: rev-C7 cells resembled their parental P19 cells in terms of morphology and cell behavior. Therefore, we chose to use C7 for further studies including susceptibility to cellular differentiation in response to RA, in vivo differentiation ability, and potential to colonize early embryos. These points have been mentioned in the revised manuscript (L307-311).

Question-3: In Fig. 4A, the EB size of P19 and C7 (KO) in d6 and d8 upon 10-8 M RA administration didn’t show big difference while 10-5 M RA on C7 showed high toxic. Did the author compare the effect of P19 and C7 in 10-7 and 10-6 M RA treatment? Please clarify this. 

Answer-3: We did not examine the effects of 10-6 M or 10-7 M RA on P19 and/or C7 clones. At present we are unable to explain why C7 cells are more sensitive to high concentrations of RA (10-5 M) compared to P19 cells. RA is a potent inhibitor of malignant tumor cell proliferation. In contrast, C7 cells can form an EB-like structure; these structures are fragile and can be easily disrupted by gentle pipetting it. We speculate that the outer components of EBs are more prone to penetration due to incomplete formation of tight adherens junctions, making C7 cells more sensitive to high RA concentrations compared to P19 cells.

Question-4: Add the molecular weight in Figure 2B.

Answer-4: As suggested, we have added the molecular weights to Figure 2B of the revised manuscript.

Question-5: Minor errors: - Fig. 2A: amend C8 to C9 according to line 243 on page 4, amend image ‘a,c’ to ‘a,d’.

Answer-5: Thank you for highlighting this point. This has been corrected in the revised manuscript.

Round 2

Reviewer 2 Report

Comments and Suggestions for Authors

The authors did a great job in addressing my comments and questions. The manuscript has been significantly improved.

Author Response

Reviewer-2 Comments and Suggestions for Authors

The authors did a great job in addressing my comments and questions. The manuscript has been significantly improved.

Thank you for your reviewing on our manuscript.

Reviewer 3 Report

Comments and Suggestions for Authors

The manuscript has been improved by the revision. However, some comments below should be addressed:

1. Figure 2B showed that α-catenin/α-tubulin ratio of Rev-C7 seemed higher than P19 cells, please provide the quantification figure and clarify this (if difference observed).

2. DAPI/Hoechst staining should be included in Figure 3B and 4B (immunofluorescence staining).

3.  Please use box to label where the Figure b&d located in Figure a&c.

4. In Figure 4D, why perform comparison between P19-10^-5 M RA (b) and C7-10^-8 M RA (c)? Not between P19-10^-8 M RA (a) and C7-10^-8 M RA (c)?

Author Response

Reviewer-3 Comments and Suggestions for Authors

The manuscript has been improved by the revision. However, some comments below should be addressed:

Question-1: Figure 2B showed that α-catenin/α-tubulin ratio of Rev-C7 seemed higher than P19 cells, please provide the quantification figure and clarify this (if difference observed).

Answer-1: As pointed out by the reviewer, α-catenin/α-tubulin ratio of Rev-C7 seemed higher than P19 cells. This is solely ascribed to the overexpression of human α-catenin in Rev-C7 after transfection with a vector conferring expression of human α-catenin gene under the strong chicken β-actin-based promoter system. We think that this overexpression will not affect the entire gene expression profile, as revealed by the microRNA array’s results (Figure 1D).

Question-2: DAPI/Hoechst staining should be included in Figure 3B and 4B (immunofluorescence staining).

Answer-2: As pointed out by the reviewer, DAPI (or Hoechst33342)-staining has been frequently used for immunostaining of fixed cells. The staining allows the identification of all types of the cells subjected to immunostaining. In another word, location of immuno-active fluorescent cells can be easily detected after co-staining with DAPI. In this case, photos of cells taken under light are not always required, because location of immune-stained cells is always associated with that of DAPI-stained cells in many of papers. Unfortunately, in our case, we did not stain the fixed cells with DAPI. Instead, we provided photos taken under light together with those taken under UV, which will also allow us to know the exact location of cells or cell aggregates. In this context, it seems appropriate to provide photos taken both under UV and light (without DAPI-stained data) for proper data presentation. 

Question-3:  Please use box to label where the Figure b&d located in Figure a&c.

Answer-3: In the case of Figure 5B, the previous Figure 5B-a,c (low-magnified photos) were not so important. Instead, we put the boxes shown in Figure 5B-b and -d in the previous text as Figure 5B-a and -b, respectively, in the re-revised text.

Question-4: In Figure 4D, why perform comparison between P19-10^-5 M RA (b) and C7-10^-8 M RA (c)? Not between P19-10^-8 M RA (a) and C7-10^-8 M RA (c)?

Answer-4: Thank you for your suggestion. We made a careless mistake. We corrected this point (please see the re-revised text).

Round 3

Reviewer 3 Report

Comments and Suggestions for Authors

The manuscript was improved after revision. No comments now.